# A Cross-Sectional Study of Attitudes toward Willingness to Use Enhancement Technologies: Implications for Technology Regulation and Ethics

**DOI:** 10.3390/biotech11030021

**Published:** 2022-06-23

**Authors:** Eisuke Nakazawa, Katsumi Mori, Makoto Udagawa, Akira Akabayashi

**Affiliations:** 1Department of Biomedical Ethics, University of Tokyo Faculty of Medicine, 7-3-1 Hongo, Bunkyo-ku, Tokyo 113-0033, Japan; nakazawa@m.u-tokyo.ac.jp (E.N.); katsumim@m.u-tokyo.ac.jp (K.M.); udagawa77@m.u-tokyo.ac.jp (M.U.); 2Division of Medical Ethics, New York University School of Medicine, 227 East 30th Street, New York, NY 10016, USA

**Keywords:** willingness, enhancement, technology regulation, ethics, empirical study, Japan

## Abstract

Neuroenhancement is rapidly re-emerging as a research topic because of the development of minimally invasive brain intervention technologies, including neurofeedback. However, public attitude toward enhancement technologies remains relatively unexplored. To fill this gap in the literature, we conducted an online survey of 1258 people in Japan who were presented with four scenarios depicting minimally and highly invasive enhancement interventions. Approximately 20% of the respondents stated that they were willing to use enhancement technologies, whereas 80% were not. Most respondents were cautious about using enhancement technologies. We used a generalized linear mixed-effects model to study the association between the type of intervention and participants’ willingness to use such technologies. Factors related to willingness to use these technologies included interventions’ degree of invasiveness, as well as participants’ gender, educational attainment, and limit or suppression experiences. We also examined the influence of others’ choices and behaviors, and participants’ tolerance toward others’ use of enhancement technologies. We explored important aspects of policymaking vis à vis enhancement technologies. This study could provide valuable insights for a debate on the ethics and regulation of enhancement technologies.

## 1. Introduction

The development of minimally invasive brain intervention technologies (e.g., neurofeedback) [1,2,3] has the potential to increase the demand for enhancement technologies among healthy adults. In this context, the term “enhancement” refers to using medical technologies beyond their therapeutic purposes, aiming to improve individuals’ cognitive abilities, skills, or outcomes [4,5]. Generally, the ethical concerns surrounding enhancement include equality, authenticity, and social coercion [6,7]. Since 2012, several studies have examined the general public’s (including students, teachers, and medical professionals) attitudes and willingness to use these technologies, such as for cognitive and emotional neuro enhancement. Although most studies have focused on pharmaceutical neuro enhancement (i.e., medication), some studies have examined non-pharmaceutical techniques (e.g., brain stimulation) [8,9,10]. The key topics in the ethical discussion on enhancement include users’ preferences and risk perceptions of enhancement technologies. Very few surveys in Japan have focused on citizens’ willingness to pursue enhancement [11]. Thus, it would be useful to determine users’ degree of willingness to pursue cognitive and emotional enhancement [12], which are generally considered the main categories of mental enhancement. Today, as minimally invasive enhancement technologies (e.g., neurofeedback) are considered useful tools, it is worth exploring users’ demand for both pharmaceutical and non-pharmaceutical enhancement using minimally invasive brain intervention technologies. Furthermore, social coercion has been noted in several prior studies on enhancement [13,14,15]; however, research on social suppression and enhancement technologies has not yet fully developed [16]. The influence of the choices and actions of others must be considered to reliably determine users’ enhancement needs.

Therefore, we surveyed the Japanese general public’s attitudes toward enhancement technologies and their regulation. We asked participants about their use of enhancement technologies, their intolerance of and conformity to the behavior of others who use enhancement technologies, as well as their views on the governmental regulation of enhancement technologies. We clarified the factors pertaining to each item and explored aspects relevant to the regulation of enhancement technologies.

## 2. Methods

### 2.1. Participants

In March 2021, we sent a survey request via email to individuals aged 20 years and above who were registered with a major internet research company in Japan. We accepted responses until we reached the sample’s planned gender and age distribution numbers. As we conducted a survey of the Japanese population’s attitudes toward enhancement technology, representativeness was prioritized. Geographically and culturally, Japan is divided into 11 major regions (Hokkaido, Tohoku, Hokuriku, Kanto, Tokai, Chubu, Kinki, Chugoku, Shikoku, Kyushu, and Okinawa). Therefore, we set a target of approximately 1200 participants, considering that approximately 100 people would be sampled from each region and that the number of participants would be even greater in regions with larger populations. Thus, we received responses from 1258 people. Consent was considered to have been given when candidates opted to answer the survey questions.

### 2.2. Survey Content

We presented the respondents with four virtual scenarios depicting the use of enhancement technologies and asked them about their attitudes toward each. The scenarios were set as one of two types of intervention: cognitive (for concentration) and emotional enhancement (for personality). Two possible intervention methods were presented: magnetic resonance imaging (MRI; minimally invasive) and drugs (highly invasive) (Table 1). For MRI scenarios, real-time functional MRI-based neurofeedback was assumed.

Neurofeedback is a novel technology with high potential for future development, while MRI is more commonly used in daily medical practice than electroencephalography or other technologies; thus, we expected that the participants in the study would have a high level of awareness of MRI neurofeedback. To improve participants’ understanding of the interventions using MRI and those using drugs, they were provided with a brief written explanation of the technology.

For each scenario, we asked the following questions:“Do you want to use this enhancement technology?” (i.e., “willingness to use enhancement technology”).“What would you do if you wanted to use this enhancement technology but others did not?” or “What would you do if you did not want to use this enhancement technology but others did?” In doing so, we effectively examined the respondent’s willingness to use these technologies and how it differs from that of others (“synchronization with other people’s usage behavior”).“Would it be permissible for others to use this technology if you were unable to use it, even if you wanted to?” (i.e., “intolerance toward use by others”).“Should this enhancement technology be regulated by the government?” (i.e., “government regulation”).

### 2.3. Limit/Suppression Experiences

In this study, the term “limit” refers to the individual’s “inability to reach a better condition despite all efforts toward achieving a given goal or purpose”, while the term “suppression“ refers to the individual “realizing what their limits are before reaching them, stopping further efforts, and compromising”. Based on these definitions, we asked the subjects whether they experienced a “limit” or “suppression” experience.

### 2.4. Participants’ Characteristics

We collected data on respondents’ demographic characteristics, including their age, gender, highest educational attainment, annual household income, and place of residence. In Japan, ordinance-designated cities are administrative districts with a relatively high population density and well-developed commercial areas. Individuals living in ordinance-designated cities were considered urban residents, while those who lived elsewhere were considered non-urban residents.

### 2.5. Analysis

We used a general linear mixed-effects model to examine the relationship among the target and means of enhancement technology interventions, participants’ demographic attributes, as well as their limit and/or suppression experience for each of the following attitudes: “willingness to use enhancement technology (wish/do not wish to use)”, “synchronization with other people’s usage behavior (synchronized/not synchronized)”, “intolerance toward use by others (not tolerated/tolerated)”, and “government regulation (should be fully banned/does not need to be fully banned)”.

Participants’ different attitudes were established as the objective variables. The intervention target (cognition/emotion) and means (minimally/highly invasive) of intervention, as well as participants’ gender, age, final academic level, annual household income, place of residence, and limit/suppression experiences were used as explanatory variables. Limit/suppression experiences were classified as follows: “limit experience present”, “limit experience absent but suppression experience present”, and “both absent”. For participants who experienced both, they were categorized into “limit experience present” group. For the objective variables of “synchronization with other people’s usage behavior”, “intolerance toward use by others”, and “government regulation”, we also added “willingness to use enhancement technology” as the explanatory variable. Each respondent engaged with all four scenarios. Individual respondents were established as variable effects in the model in considering intra-individual correlations. Statistical analysis was conducted using SAS software, version 9.4 (SAS Institute Inc., Cary, NC, USA), while *p* < 0.05 was considered statistically significant.

### 2.6. Ethical Considerations

This study received approval from the Research Ethics Committee of the Faculty of Medicine of the University of Tokyo (2020355NI). 

## 3. Results

Table 2 presents the attributes of the respondents and their limit and suppression experiences. In terms of gender, participants were sampled so that men and women were almost equal in number. Sampling was conducted such that age groups of every 10 years over the age of 20 years were almost equal in number. However, the analyses used the classifications of 20–39 years, 40–64 years, and 65 years and above. Approximately 32% of the respondents had a limit experience present or absent but a suppression experience present, whereas both were absent for about 36% of respondents.

Table 3 shows the aggregated results of attitudes toward the use of enhancement technologies. About 30% of respondents indicated a willingness to use enhancement technologies in scenarios A and C (which refer to interventions using MRI). By contrast, between 23% and 24% of participants indicated a willingness to use enhancement technologies in scenarios B and D (which refer to interventions using highly invasive drugs). Respondents’ synchronization with others’ usage behavior was lower for scenarios B and D compared with scenarios A and C. The differences owing to the means of intervention were clear. Respondents’ intolerance toward usage by others was slightly above 21% for scenarios A, B, and D and slightly lower for scenario C (19.6%). However, there were hardly any differences among the scenarios. Around 18% of respondents indicated that enhancement technologies should be fully banned for scenarios A and C, while 20–21% indicated the same for scenarios B and D. There were differences according to the means of intervention here, as well.

Table 4 presents the results from a linear mixed model with respondents’ attitude toward the use of enhancement technologies as the objective variable. There were statistically significant associations between respondents’ willingness to use enhancement technology and the means of intervention, gender, highest educational attainment, and limit experience. Male respondents, those with higher educational attainment, and those with prior limit experiences indicated a higher preference for minimally invasive interventions. Only willingness to use enhancement technologies was statistically significantly correlated (adjusted odds ratio [AOR] = 7.994) with synchronization, which indicated that peer pressure had a considerable impact. Gender and age were statistically significantly associated with intolerance toward use by others; men and younger respondents were less tolerant if they were not able to use enhancement technologies themselves. There were no significant correlations for either the target or the means of intervention, while respondents’ intolerance toward use by others was associated more with personal attributes than with the nature of the enhancement itself. There were statistically significant correlations between government regulation and the means of intervention, highest educational attainment, place of residence, and willingness to use enhancement technologies. More respondents indicated that enhancement technologies should be banned by the government for highly invasive drug interventions compared with minimally invasive MRI interventions. Additionally, individuals with moderate educational attainment, urban residents, and those with low willingness to use enhancement technologies were more likely to indicate that such technologies should be banned.

## 4. Discussion

This is the first study to comprehensively investigate the Japanese population’s attitudes toward the use of enhancement technologies. First, Table 3 shows that respondents’ willingness to use enhancement technologies in the four scenarios was only 20–30%. This shows that most people do not wish to use enhancement technologies in Japan. Similarly, a survey on attitudes toward pharmaceutical cognitive enhancement among Japanese students found that 68.6–72.0% (2017–2019, biannual survey) did not want to use pharmaceuticals for cognitive enhancement [11]. Although direct international comparisons are difficult, a survey of Swiss students’ attitudes showed that 57.9% of non-cognitive enhancement users were willing to try a product to increase their intellectual quotient without side effects, while 90.6% of cognitive enhancement users indicated the same [17].

Approximately 80% of the respondents remained consistent in their attitudes and did not engage in synchronization with others. This suggests that participants who did not want to use these technologies were less likely to respond differently regardless of what other people did. Similarly, approximately 80% were tolerant toward others’ use of enhancement technologies. Additionally, approximately 80% indicated that there was no need for banning such technologies. In sum, most respondents thought it acceptable for others to use enhancement, did not mind others’ behavior, and did not find a full ban by the government was needed.

The results of the general linear mixed-effects model in Table 4 show that the items presented in the results were significantly correlated. Respondents’ willingness to use enhancement technology was low for highly invasive interventions. Additionally, highly educated people were more willing to use enhancement technology compared with their less educated counterparts. This suggests that individuals with higher education may understand enhancement better or may require more abilities than those they currently possess. The low willingness to use enhancement technologies among individuals without limit experiences may be explained by the fact that they did not feel like they had reached their limit (cognitively or emotionally) and thus had no need for enhancement.

Most respondents aligned with others who did not use enhancement technology, even if they themselves wanted to use it. We speculate that this indicates Japanese people’s concerns regarding interventions’ effects and safety. Given the cultural aspects of life in Japan, it may also reflect the fact that Japan is a homogeneous and collectivist society (i.e., that the individual is considered part of the group, and group cohesion is prioritized over individuality); therefore, someone doing something different or standing out from others may result in that individual being socially sanctioned, thereby resulting in them refraining from doing so.

Some individuals may think that regulation is necessary for highly invasive interventions, while those wanting to receive enhancement may think that regulations would decrease the opportunities to access enhancement technologies. In this study, urban respondents thought that regulation was unnecessary. Finally, the willingness to use enhancement technologies was low among women, whereas tolerance toward others’ usage was low among men. This could suggest that women remained skeptical of these technologies and did not wish to engage in such interventions but were still tolerant and kind toward others who were willing to use such technologies. This could suggest that women are both cautious and thoughtful regarding the use of enhancement technologies.

### 4.1. Policy Implications

(1)There was an agreement, to some extent, that government regulation is necessary for highly invasive interventions if enhancement technologies are to be used.(2)Impact of one’s surroundings: Most respondents aligned with others who did not use enhancement technologies, even if they wanted to use them. Thus, the behaviors of others had an effect.

These two points must be carefully considered while formulating policies toward regulating the use of enhancement technologies in the future.

### 4.2. Limitations

(1)This study was an online survey; thus, the representative characteristics of the respondents are unknown.(2)The scenarios used were not well validated.(3)There is no information on how accurately the participants understood the scenarios.(4)A high percentage of respondents did not reveal their annual household income (20.8%), so it is unclear how income affected the findings.(5)The use of dichotomized responses might have led to a loss of information.(6)Some important variables (e.g., previous experience with enhancement technologies, religiosity, and technophilia vs. technophobia) may have been overlooked.(7)Lastly, this study used a cross-sectional design. One key limitation of cross-sectional studies is that the temporal pre- and post-relationships between items are unclear; thus, it is not possible to discuss causal relationships. However, in this study, all the items we included have a clear direction of causality, if any. Only one item, “desire to use enhancement technologies”, has neither a clear temporal relationship nor a clear causal direction with “attitudes toward other enhancement technologies”. Lastly, the discussion was limited to this study’s relevance.

## 5. Conclusions

We conducted a survey of the general Japanese public to understand their attitudes toward the use of enhancement technologies. Most people were cautious about the use of such technologies. By using general linear mixed-effects models, factors that are associated with those attitudes were analyzed.

In sum, this survey’s respondents did not mind if others used enhancement technologies and did not consider it necessary for the government to ban them. Understanding the cultural environment (extremely strongly correlated at AOR = 7.994) is important for policymaking vis à vis enhancement technologies, not only in Japan but worldwide.

## Figures and Tables

**Table 1 biotech-11-00021-t001:** Scenarios depicting the use of enhancement technologies.

Scenario	Intervention Target	Intervention Means	Scenario Summary
A	Cognition (concentration)	Minimally invasive (MRI)	Improve concentration by conducting simple training under MRI before studying for an important qualification examination. Training for 2 h × 4 days necessary. Almost no side effects. Costs approximately JPY 40,000.
B	Cognition (concentration)	Highly invasive (drug)	Improve concentration by taking drugs typically used to treat a specific illness before studying for an important qualification examination. Taken daily until test. Possibility of mild side effects. Costs approximately JPY 10,000.
C	Emotion (personality)	Minimally invasive (MRI)	Improve positivity and cheerfulness to make a good impression at a job interview by conducting simple training under MRI. Training for 2 h × 4 days necessary. Few side effects. Costs approximately JPY 40,000.
D	Emotion (personality)	Highly invasive (drug)	Improve positivity and cheerfulness to make a good impression at a job interview by taking drugs typically used to treat specific illnesses. Take daily until the test. Possibility of mild side effects. Costs approximately JPY 10,000.

**Table 2 biotech-11-00021-t002:** Respondents’ attributes and limit/suppression experiences.

		*n*	(%)
Gender	Male	618	(49.1%)
	Female	640	(50.9%)

Age (years)	20–39	403	(32.0%)
	40–64	551	(43.8%)
	>65	304	(24.2%)

Highest educational attainment	Junior high school/high school	414	(32.9%)
	Vocational school, junior and technical college	276	(21.9%)
	University/graduate school	556	(44.2%)
	Do not wish to answer	12	(1.0%)

Household income	<JPY 4 million	432	(34.3%)
	JPY 4–8 million	377	(30.0%)
	>JPY 8 million	187	(14.9%)
	Do not know/do not wish to answer	262	(20.8%)

Residence	Non-urban area	802	(63.8%)
	Urban area	456	(36.2%)
Limit/suppression experience	Limit experience present	401	(31.9%)
	Limit experience absent but suppression experience present	403	(32.0%)
	Both absent	454	(36.1%)

**Table 3 biotech-11-00021-t003:** Attitude toward the use of enhancement technology.

	Scenario A		Scenario B		Scenario C		Scenario D	
Willingness to use enhancement technology								
Wish to use	374	(29.7%)	293	(23.3%)	378	(30.0%)	297	(23.6%)
Do not wish to use	884	(70.3%)	965	(76.7%)	880	(70.0%)	961	(76.4%)
Synchronization with other people’s usage behavior								
Synchronize	268	(21.3%)	238	(18.9%)	267	(21.2%)	238	(18.9%)
Do not synchronize	990	(78.7%)	1020	(81.1%)	991	(78.8%)	1020	(81.1%)
Intolerance toward use by others								
Cannot tolerate	273	(21.7%)	273	(21.7%)	246	(19.6%)	269	(21.4%)
Can tolerate	985	(78.3%)	985	(78.3%)	1012	(80.4%)	989	(78.6%)
Government regulation								
Should be fully banned	238	(18.9%)	270	(21.5%)	225	(17.9%)	252	(20.0%)
Does not need to be fully banned	1020	(81.1%)	988	(78.5%)	1033	(82.1%)	1006	(80.0%)

**Table 4 biotech-11-00021-t004:** Factors related to attitudes toward the use of enhancement technologies.

a. Factors related to attitudes toward the use of enhancement technologies (*n* = 993)							
		Willingness to use enhancement technology			Synchronization with other people’s usage behavior		
		AOR (95% CI)		*p*-Value	AOR (95% CI)		*p*-Value
Intervention target	Cognitive (concentration)	-			-		
	Emotional (personality)	0.988	(0.825, 1.182)	0.891	0.902	(0.741, 1.097)	0.301
Intervention means	Minimally invasive (MRI)	-			-		
	Highly invasive (drugs)	0.612	(0.511, 0.734)	<0.001	1.010	(0.829, 1.231)	0.922
Gender	Male	-			-		
	Female	0.573	(0.410, 0.800)	0.001	1.194	(0.876, 1.628)	0.262
Age (years)	20–39	-			-		
	40–64	0.762	(0.525, 1.106)	0.153	0.752	(0.532, 1.062)	0.105
	>65	0.765	(0.495, 1.181)	0.226	0.862	(0.578, 1.287)	0.469
Highest educational attainment	Junior high/high and	-			-		
	vocational school, junior and technical college	1.524	(0.962, 2.412)	0.072	1.193	(0.788, 1.807)	0.404
	University/graduate school	1.938	(1.323, 2.837)	0.001	0.826	(0.578, 1.179)	0.291
Household income (JPY)	<4 million	-			-		
	4–8 million	1.134	(0.792, 1.623)	0.492	1.368	(0.981, 1.908)	0.065
	>8 million	1.096	(0.694, 1.729)	0.695	1.482	(0.970, 2.263)	0.069
Residence	Non-urban	-			-		
	Urban	1.124	(0.807, 1.566)	0.489	1.056	(0.777, 1.434)	0.729
Limit/suppression experience	Limit experience present	-			-		
	Suppression experience only	0.718	(0.487, 1.058)	0.094	0.776	(0.542, 1.110)	0.165
	Both absent	0.611	(0.412, 0.908)	0.015	0.726	(0.504, 1.046)	0.086
Willingness to use	Want to use				-		
	Do not want to use				7.994	(6.203, 10.302)	<0.001
b. Factors related to attitudes toward the use of enhancement technologies (*n* = 993)							
		Intolerance toward use by others			Government regulation		
		AOR (95% CI)		*p*-Value	AOR (95% CI)		*p*-Value
Intervention target	Cognitive (concentration)	-			-		
	Emotional (personality)	0.878	(0.717, 1.077)	0.212	0.866	(0.701, 1.069)	0.180
Intervention means	Minimally invasive (MRI)	-			-		
	Highly invasive (drugs)	1.206	(0.983, 1.478)	0.072	1.270	(1.027, 1.570)	0.027
Gender	Male	-			-		
	Female	0.510	(0.356, 0.732)	0.000	1.204	(0.829, 1.749)	0.329
Age (years)	20–39	-			-		
	40–64	0.667	(0.450, 0.987)	0.043	1.010	(0.661, 1.545)	0.962
	>65	0.340	(0.209, 0.551)	<0.001	1.426	(0.877, 2.317)	0.152
Final educational background	Junior high/high and	-			-		
	vocational school, junior and technical college	1.606	(0.987, 2.614)	0.057	1.748	(1.065, 2.868)	0.027
	University/graduate school	1.087	(0.720, 1.640)	0.693	1.214	(0.791, 1.864)	0.375
Household income (JPY)	<4 million	-			-		
	4–8 million	1.114	(0.758, 1.636)	0.584	0.937	(0.627, 1.400)	0.751
	>8 million	1.176	(0.720, 1.920)	0.517	1.269	(0.761, 2.117)	0.361
Residence	Non-urban	-			-		
	Urban	0.853	(0.595, 1.224)	0.388	0.664	(0.455, 0.969)	0.034
Limit/suppression experience	Limit experience present	-			-		
	Suppression experience only	0.819	(0.537, 1.249)	0.354	0.892	(0.573, 1.390)	0.613
	Both absent	1.025	(0.672, 1.562)	0.910	0.982	(0.632, 1.525)	0.935
Willingness to use	Want to use	-			-		
	Do not want to use	0.819	(0.604, 1.111)	0.199	0.310	(0.216, 0.446)	<0.001

Note: AOR = adjusted odds ratio; CI = confidence interval.

## Data Availability

Data available on request due to privacy restrictions.

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
