# Peer review of "A Cross-Sectional Study of Attitudes toward Willingness to Use Enhancement Technologies: Implications for Technology Regulation and Ethics"

_biotech, 2022, doi:10.3390/biotech11030021_

Round 1
Reviewer 1 Report
Evaluation Biotech 1774144
This online survey study, investigates the willingness to use enhancement 198 technology in the Japanese adult population (N for the surveyed sample > 1200) as a function of type of technological invasiveness (MRI vs drugs) and type of targeted function (cognition vs. Emotion). The main results showed that the majority of respondents did not want to use enhancement technology but tolerated if others used it and did not think that it would be necessary a ban by the government especially for techniques with low-invasiveness. Although very limited in scope, this survey is interesting and useful for the literature. I have some concerns that I would like the authors to address if possible.
Minor points
- Page 2: “Furthermore, the needs of user enhancement are 53 thought to be influenced by others and their behaviors.” A reference to back up this assertion is needed here.
- Page 2: “We accepted 67 responses until we reached the planned numbers in terms of gender and age.” Which were the planned numbers and why (e.g., Power analysis...)?
- The authors lightly touched the cultural factors (this survey was carried out in Japan), however the issue generalizability of the results of the reported survey in other cultures should be further expanded.
- Page 2: The authors should explain to the reader what type of MRI neurofeedback they meant (e.g., functional MRI based neurofeedback?) and whether this was clear to the interviewed individuals as well. The rationale for focusing on such a technique should also be made clear to the reader. Some other techniques might have been also appropriate to test as scenarios, since their use could be perceived as less expensive or more comfortable. Some examples follow: EEG-based Neurofeedback or adaptive computer-based cognitive or emotional interventions/training which have been used and validated for both healthy individuals (e.g., Gruzelier, 2014, https://doi.org/10.1016/j.neubiorev.2013.09.015) and patients with brain damage (e.g., Tarantino et al., 2021, https://doi.org/10.3390/brainsci11081002). A note on this issue (why MRI was chosen for the scenarios?) is desirable.
- Table 4 needs to be better formatted (e.g., digits belonging to the same value, e.g., p values, should preferably fit in the same line) for a better readability.
- Page 8, Limit N. 7: “7) Lastly, this study took the cross-sectional design. The limit of cross-sectional de-246 sign should be fully recognized.” This limit should be fully elaborated.
Minor slips:
- Page 2, first line: “these ethical discussion”, please put an s after “discussion”
- Page 2: the authors should spell out “MRI” the first time this acronym occurs.
Author Response
Responses to Reviewer 1
Thank you so much for your useful comments.
- Page 2: “Furthermore, the needs of user enhancement are thought to be influenced by others and their behaviors.” A reference to back up this assertion is needed here.
Thank you for your comment. Since our description is unclear, we would like to change the sentences you mentioned as follows:
Furthermore, the needs of user enhancement are thought to be influenced by others and their behaviors. Emotional enhancement is closely associated with morality [13]. Determining the needs of user enhancement reliably requires considering the influences of the choices and actions of others on the user.
[13]Greene, J. D., Sommerville, R. B., Nystrom, L. E., Darley, J. M., & Cohen, J. D. (2001). An fMRI investigation of emotional engagement in moral judgment. Science, 293(5537), 2105–2108.
→→→
Furthermore, the social coercion of enhancement has already been noted in several studies [13–15]; however, research on the social suppression of enhancement has not yet fully progressed. In particular, the social perspective is significant because emotional enhancement is closely associated with morality [16]. The influence of the choices and actions of others must be considered to reliably determine user enhancement needs.
[13] Bell, S., Partridge, B., Lucke, J., & Hall, W. (2001). Australian University students’ attitudes towards the acceptability and regulation of pharmaceuticals to improve academic performance. Neuroethics 6, 197–205.
[14] Racine, E., and C. Forlini. 2010. Cognitive enhancement, lifestyle choice or misuse of prescription drugs?: Ethics blind spots in current debates. Neuroethics 3 (1), 1–4.
[15] Claire T. D., Humphries S., & Chatterjee, A. (2020). Public opinion on cognitive enhancement varies across different situations, AJOB Neuroscience 11:4, 224-237.
[16] Greene, J. D., Sommerville, R. B., Nystrom, L. E., Darley, J. M., & Cohen, J. D. (2001). An fMRI investigation of emotional engagement in moral judgment. Science, 293(5537), 2105–2108.
- Page 2: “We accepted responses until we reached the planned numbers in terms of gender and age.” Which were the planned numbers and why (e.g., Power analysis...)?
As this study was conducted as a survey of Japanese attitudes toward enhancement technology, the number of participants was set to ensure a certain level of representativeness. Geographically and culturally, Japan is divided into 11 major regions (Hokkaido, Tohoku, Hokuriku, Kanto, Tokai, Chubu, Kinki, Chugoku, Shikoku, Kyushu, and Okinawa). We set a target number of approximately 1,200 participants, considering that about 100 people would be sampled from each region and that the number of participants will be even greater in regions with larger populations.
- The authors lightly touched the cultural factors (this survey was carried out in Japan), however the issue generalizability of the results of the reported survey in other cultures should be further expanded.
It is difficult to compare questionnaire surveys on attitudes toward enhancement because the definition of enhancement (i.e., the techniques used for enhancement, especially the definition of drugs) cannot be clearly defined in the same manner for each culture. Thus, we expect that East Asian people may have a stronger collective mindset and a relatively lower desire for enhancement than their Western counterparts because East Asian people share a Confucian culture derived from the Chinese tradition. Although only a partial review, we have added the results of a questionnaire survey on attitudes conducted in Switzerland to the discussion. This survey of attitudes among students in Switzerland showed that, even among non-cognitive enhancement users, 57.9% would be prepared to try a product that has no side effects to increase their IQ.
This result is similar to the findings of our study, indicating that Japanese people are cautious in their attitudes toward enhancement. Although simple international comparisons are difficult, a survey of attitudes among students in Switzerland showed that, even among non-cognitive enhancement users, 57.9% would be prepared to try a product that has no side effects to increase their IQ (cognitive enhancement user: 90.6%) [17].
[17] Ott R, Biller-Andorno N. Neuroenhancement among Swiss students--a comparison of users and non-users Pharmacopsychiatry. 2014;47(1):22-28. doi:10.1055/s-0033-1358682
- Page 2: The authors should explain to the reader what type of MRI neurofeedback they meant (e.g., functional MRI based neurofeedback?) and whether this was clear to the interviewed individuals as well. The rationale for focusing on such a technique should also be made clear to the reader. Some other techniques might have been also appropriate to test as scenarios, since their use could be perceived as less expensive or more comfortable. Some examples follow: EEG-based Neurofeedback or adaptive computer-based cognitive or emotional interventions/training which have been used and validated for both healthy individuals (e.g., Gruzelier, 2014, https://doi.org/10.1016/j.neubiorev.2013.09.015) and patients with brain damage (e.g., Tarantino et al., 2021, https://doi.org/10.3390/brainsci11081002). A note on this issue (why MRI was chosen for the scenarios?) is desirable.
As intervention using MRI, we assume decoded neurofeedback using fMRI. We chose to use fMRI-based neurofeedback instead of EEG-based neurofeedback because the former is more frequently used in daily medical practice, including physical examinations, and is more widely recognized by the public than EEG. MRI-based neurofeedback was also chosen because it does not require a specified region of interests, is highly versatile, and has great potential for future development. To give the research participants an overview of MRI neurofeedback, we used the following plain language.
[Ability-enhancement using MRI and drugs]
MRI, which can record images of the brain, is a familiar medical device used in brain scans. MRIs are usually used to detect brain diseases, but recently, it has become possible to brighten one's mood and improve one's ability to concentrate by performing simple training while in the MRI. Similarly, certain drugs can be prescribed to patients to brighten their mood and improve concentration. In addition to treating diseases or disorders, these MRI techniques and medications can also be used on healthy people. A brighter mood can help people interact more pleasantly; hence, they may perform better in job interviews. Likewise, among students, improved concentration may promote more efficient studying.
Thus, we added the explanation after the sentence, “The scenarios were set as combinations of the two types of intervention subjects—concentration-focused intervention (cognition) and personality-focused intervention (emotion)—and the two methods of intervention using MRI (minimally invasive) and intervention using drugs (highly invasive) (Table 1)” as follows:
As an intervention using MRI, real-time functional MRI-based neurofeedback was assumed.
It is a novel technology with high potential for future development, and MRI is more commonly used in daily medical practice than EEG or other technologies, so we expected that the participants in the study would have a high level of awareness of MRI neurofeedback. To improve participants' understanding of both interventions using MRI and those using drugs, they were provided a brief written explanation of the technology.
- Table 4 needs to be better formatted (e.g., digits belonging to the same value, e.g., p values, should preferably fit in the same line) for a better readability.
Thank you for the advice. We have divided the table into two parts to make it easier to read.
- Page 8, Limit N. 7: “7) Lastly, this study took the cross-sectional design. The limit of cross-sectional design should be fully recognized.” This limit should be fully elaborated.
The limitation of a cross-sectional study is that the temporal pre- and post-relationships between items are unclear; thus, it is not possible to discuss causal relationships. In this study, however, all the items we deal with have a clear direction of causality, if any. Only one item, "desire to use enhancement technology," has neither a clear temporal relationship nor a clear causal direction with attitudes toward other enhancement technologies, and the discussion is limited to a discussion of relevance only.
Minor slips:
- Page 2, first line: “these ethical discussion”, please put an s after “discussion”
Thank you so much.
- Page 2: the authors should spell out “MRI” the first time this acronym occurs.
Thank you so much.
Once again, thank you very much for your helpful comments. We are ready to revise the manuscript further if you so desire.

Reviewer 2 Report
Dear authors,
I congratulate you on the article, which is interesting and the study is well conducted. However, there are numerous issues that have to be solved, mostly regarding the structure of the article. There are many incorrect/unclear statements in the article (only a few examples are given below).
- line 35 - the word need is used improperly here. Maybe requests?
- line 48-49 - the main type of enhancement, at least now, is physical (appearance)
- line 50-53 - the sentence does not seem to make sense. Please rephrase
- line 57 - the wording is unclear. What are basic factors?
- line 66 - what are monitors?
- Table 1 - it seems that scenario D is incorrect
Also, the discussion section only discusses the results of the study, without any correlation with other studies from the scientific literature. Overall, the number of references is very low, even though both cognitive and emotional enhancement are very highly debated topic in medical ethics in the last two decades.
Author Response
Responses to Reviewer 2
Thank you so much for your useful comments.
- line 35 - the word need is used improperly here. Maybe requests?
Thank you for your correction.
The word “requests” is better than the word “need.”
- line 48-49 - the main type of enhancement, at least now, is physical (appearance)
Thank you. We took it for granted and instead lost sight of it.
We will change:
“It is probably useful to determine the degree of user willingness to cognitive and emotional enhancement [12], which are generally considered the main categories of enhancement.”
The revision is:
“It is probably useful to determine the degree of user willingness to pursue cognitive and emotional enhancement [12], which are generally considered the main categories of mental-state enhancement.”
- line 50-53 - the sentence does not seem to make sense. Please rephrase
Thank you.
As we mentioned above, we changed some sentences as follows:
Furthermore, the needs of user enhancement are thought to be influenced by others and their behaviors. Emotional enhancement is closely associated with morality [13]. Determining the needs of user enhancement reliably requires considering the influences of the choices and actions of others on the user.
[13]Greene, J. D., Sommerville, R. B., Nystrom, L. E., Darley, J. M., & Cohen, J. D. (2001). An fMRI investigation of emotional engagement in moral judgment. Science, 293(5537), 2105–2108.
→→→
Furthermore, the social coercion of enhancement has already been noted in several studies [13–15]; however, research on the social suppression of enhancement has not yet fully progressed. In particular, the social perspective is significant because emotional enhancement is closely associated with morality [16]. The influence of the choices and actions of others must be considered to reliably determine user enhancement needs.
[13] Bell, S., Partridge, B., Lucke, J., & Hall, W. (2001). Australian university students’ attitudes towards the acceptability and regulation of pharmaceuticals to improve academic performance. Neuroethics 6, 197–205.
[14] Racine, E., and C. Forlini. 2010. Cognitive enhancement, lifestyle choice or misuse of prescription drugs?: Ethics blind spots in current debates. Neuroethics 3 (1), 1–4.
[15] Claire T. D., Humphries S., & Chatterjee, A. (2020). Public Opinion on Cognitive Enhancement Varies across Different Situations, AJOB Neuroscience 11:4, 224-237.
[16] Greene, J. D., Sommerville, R. B., Nystrom, L. E., Darley, J. M., & Cohen, J. D. (2001). An fMRI investigation of emotional engagement in moral judgment. Science, 293(5537), 2105–2108.
- line 57 - the wording is unclear. What are basic factors?
Thank you. We changed the sentence, “Therefore, the objective of this research is to obtain basic factors when considering the enhancement technology, namely, the ethics in enhancement technology as well as enhance technology regulation.”
to
“Therefore, we surveyed the attitudes of the Japanese general public to obtain basic data describing their views on the use of enhancement technology and its regulation.”
- line 66 - what are monitors?
Thank you. We changed the word “monitors” to “research participants.”
- Table 1 - it seems that scenario D is incorrect
We apologize for the error.
We have now changed
“Improve positivity and cheerfulness by taking drugs that are used to treat a certain illness to study for important qualification examinations. Taken daily until test. Possibility of mild side effects. Costs approximately 10,000 yen.”
to
“Improve positivity and cheerfulness to create a good impression at a job interview by taking medications traditionally used for treating certain illnesses. Take daily until the test. Possibility of mild side effects. Costs approximately 10,000 yen.”
Also, the discussion section only discusses the results of the study, without any correlation with other studies from the scientific literature. Overall, the number of references is very low, even though both cognitive and emotional enhancement are very highly debated topic in medical ethics in the last two decades.
Thank you very much for this advice. We have added some additional references. We also added the following correlations to the Discussion.
A survey on pharmaceutical cognitive enhancement attitudes among Japanese students found that 68.6–72.0% (2017–2019, biannual survey) answered "no" to the question "do you want to use pharmaceuticals" [11]. This result is similar to that of our study, indicating that the Japanese are cautious in their attitudes toward enhancement. Although simple international comparisons are difficult, a survey of attitudes among students in Switzerland showed that, even among non-cognitive enhancement users, 57.9% would be prepared to try a product that has no side effects to increase their IQ (cognitive enhancement user: 90.6%) [17].
[17] Ott R, Biller-Andorno N. Neuroenhancement among Swiss students--a comparison of users and non-users Pharmacopsychiatry. 2014;47(1):22-28. doi:10.1055/s-0033-1358682
Once again, thank you very much for your helpful comments. We are ready to revise the manuscript further if you so desire.
Round 2
Reviewer 2 Report
The syntax is still difficult to understand in some places of the article. A professional proofreading should be done before finally accepting the article
Author Response
Dear the Reviewer 3,
Thank you for your comment. We have asked a professional proofreading company, and have attached Certification of proofreading from that company.
We hope English is now OK, an will be publishable.
Best,
Akira Akabayashi